# Does a Healthy Lifestyle Lower the Elevated Risk of Obesity Caused by Caesarian Section Delivery in Children and Adolescents?

**DOI:** 10.3390/nu14173528

**Published:** 2022-08-26

**Authors:** Rong Lian, Zheng-He Wang, Zhi-Yong Zou, Yan-Hui Dong, Yi-De Yang, Jun Ma

**Affiliations:** 1Guangzhou First People’s Hospital, School of Medicine, South China University of Technology, Guangzhou 510180, China; 2Department of Microbiology, Sun Yat-sen University Zhongshan School of Medicine, Guangzhou 510080, China; 3Department of Epidemiology, School of Public Health, Southern Medical University, Guangzhou 510515, China; 4School of Public Health & Institute of Child and Adolescent Health, Peking University, Beijing 100871, China; 5Key Laboratory of Molecular Epidemiology of Hunan Province, School of Medicine, Hunan Normal University, Changsha 410081, China

**Keywords:** delivery, obesity, caesarean, lifestyle, child health

## Abstract

Background: Both caesarean section (CS) and lifestyle were linked with child adiposity. This study aimed to investigate whether CS delivery is linked with elevated risk of child adiposity regardless of a healthy lifestyle. Methods: All the subjects in this study came from a baseline survey of a national school-based program on healthy lifestyle interventions against adiposity among Chinese children and adolescents. A questionnaire was used to collect the information on delivery mode and lifestyle. According to the weighted lifestyle score, subjects were categorized into healthy, intermediate, and unhealthy lifestyle. Results: A total of 44,961 children aged 6–18 years were enrolled in the current study. Overall, 41.9% (18,855/44,961) of children were delivered by CS. Compared with children delivered by vaginal delivery, children delivered by CS had a higher adiposity risk (*OR* = 1.56; 95%*CI*: 1.46–1.66; *p* < 0.001) after adjustment for age, sex, region, mother adiposity, ethnicity, and weighted lifestyle factors. Compared with children with a healthy lifestyle, children with an unhealthy lifestyle had a higher risk of child adiposity (*OR* = 1.31; 95%*CI*: 1.19–1.44). Children delivered by CS who had an unhealthy lifestyle had a 106% higher (*OR* = 2.06; 95%*CI*: 1.79–2.37) risk of child adiposity compared with children delivered by vaginal delivery who had a healthy lifestyle. However, keeping a healthy lifestyle in later life seems not to offset the increased risk of child adiposity caused by CS (*OR* = 1.59; 95%*CI*: 1.39–1.82). Conclusions: Both CS and unhealthy lifestyle were linked with child adiposity risk. Keeping a healthy lifestyle did not counteract the elevated risk of child adiposity caused by CS.

## 1. Introduction

Caesarean sections (CS) are the most frequently performed surgeries among women, and CS rates are increasing in most countries across the world [1,2,3,4]. In view of the World Health Organization statement in 1985 that “regional CS rates should not exceed 10~15%”, the CS rates are too high in the majority of countries or regions. Latest data (2010–2018) from 154 countries covering 94.5% of global live births show that 21.1% of women delivered by CS worldwide, but the prevalence differs between different countries and regions [4]. The CS rate is less than 3% in western Africa; however, CS rates are especially high in many countries and regions, such as the Dominican Republic (56.4%), and Brazil (55.6%). The CS rates are about 25% to 35% in many European countries, New Zealand, and the United States [2]. In China, the CS rate was 36.2% in 2011 [2].

Undoubtedly, CS is a life-saving surgical procedure that has dramatically decreased maternal and neonatal mortality rates during the 20th century [5]. Despite this clearly positive effect, the rapid increase in CS rates worldwide has become a growing public health challenge [6,7]. Previously, a meta-analysis showed that children delivered by CS had a significantly elevated 29% risk of child adiposity, even after adjusting for maternal pre-pregnancy weight [8].

Child obesity has become a major general health problem around the world, notably in developing and poor nations [9]. In the United States, adiposity prevalence among children and adolescents was 16.9% (95%*CI*, 14.9–19.2%) [10]. A national large-scale epidemiological survey showed that the overweight and obesity rate of Chinese students aged 7–18 years was 20.4% [11]. Childhood obesity has a documented adverse effect on physical and mental health [12]. Childhood obesity is more likely to lead to non-communicable diseases at a more youthful age [13,14,15,16]. A previous study conducted by our research group found that students with obesity were associated with an 86% higher risk of IFG [16].

Apart from CS being associated with childhood obesity, it has been documented that lifestyle is associated with childhood obesity. Keeping a healthy lifestyle meant a lower childhood obesity risk [17,18,19]. Currently, several research studies have integrated diet, sleeping, physical activity, and screen time to compute a lifestyle score to determine the effect of lifestyle on health outcomes [20]. In addition, factors generally do not lead to adiposity alone, they work together complexly, interact, or have combined effects to affect the risk of adiposity. Thus, we speculated that the elevated childhood obesity risk caused by CS might be counteracted by keeping a healthy lifestyle.

Therefore, the current study aimed to determine the hypothesis that keeping a healthy lifestyle may counteract the elevated childhood obesity risk caused by CS, based on the baseline survey data from a nationwide school-based project on healthy lifestyle interventions against obesity in Chinese children aged 6–18 years.

## 2. Materials and Methods

All the participants in this study were selected from a baseline survey of a nationwide school-based project on healthy lifestyle interventions against obesity in Chinese children. The Ethical Committee of the Peking University Medical Science Center had approved this project (IRB00001052-13034), and all the participants and their parents provided informed consents voluntarily. 

### 2.1. Participants

The baseline survey of the nationwide school-based project on healthy lifestyle interventions against obesity in Chinese children aged 6–18 years was carried out in 2013. The survey involved more than 65,347 students aged 6–18 years from seven provinces or municipalities (Tianjin, Shanghai, Chongqing, Ningxia, Guangdong, Hunan, and Liaoning). A multistage sampling method was used to select the participants. About 12–16 primary and secondary schools, with a total of about 10,000 students aged 6–18 years, were enrolled in each province or municipality. The design details have been specified in a previous publication [21]. In the current study, we included 44,961 participants aged 6–18 years with complete data in the final analysis, after excluding 12,349 students without delivery mode data, 2324 students without BMI data, 3947 students with premature delivery, and 1766 students with inconsistent delivery mode between baseline and 6 months’ follow-up.

### 2.2. Data Collection and Questionnaire Survey

Data on students were collected by a standard questionnaire completed by students and one of their guardians. Information on common food intake, sleep, physical activity, and screen time was reported by the students themselves. Nevertheless, demographic information, feeding, parental education level, delivery mode, birth weight, gestation age, family history of obesity, and parent BMI were reported by their parents. 

The intake of food, such as vegetables, fruits, meats, milk, sugar-sweetened drinks, fried foods, and desserts, was collected from students using a 7-day food frequency questionnaire. The average intake of a food over the past week was calculated by the number of days multiplied by the number of servings per day and then divided by 7 [22]. The Global Physical Activity Questionnaire was used to assess each respondent’s level of physical activity [23]. 

### 2.3. Anthropological Measurements

Height and weight were determined by trained investigators following a standardized procedure. A portable stadiometer (model TZG, Bengbu, China) was used to assess the height, and a lever-type weight scale (model RGT-140, Bengbu, China) was used to measure the body weight. 

### 2.4. Definition of Obesity 

The obesity of students aged 6–18 years was defined by the Chinese body mass index (BMI) percentile criterion. BMI ≥ 95th percentile was used as the cut-off for determining obesity [24]. Mother obesity was defined based on maternal BMI at the time of the survey. BMI ≥ 28.0 kg/m^2^ was used as the cut-off for determining obesity.

### 2.5. Delivery Mode

The information on delivery mode was obtained through a guardian questionnaire. Of these, 81.1% of guardians reported delivery mode of their baby based on the medical certificate of birth. For those without a birth certificate, we required their guardians to report the delivery mode of their baby. In addition, we repeated this survey six months later and observed that the difference in delivery mode between the two surveys was less than 10%. After excluding the students with inconsistent information for reported delivery mode, all the students were classified into CS and vaginal delivery. 

### 2.6. Healthy Lifestyle Score

Four main obesity-related factors (diet, physical activity, sleeping, and screen time) were used to calculate a healthy lifestyle score [25,26,27]. These obesity-related factors were collected from a baseline survey using a standardized questionnaire. Sleep duration was defined as adequate sleep and inadequate sleep according to the Notice of the Chinese General Office of the Ministry of Education on Further Strengthening the Sleep Management of Primary and Middle School Students. The detailed method of healthy lifestyle score calculation has been described in our previous study [28]. Briefly, the weighted standardized lifestyle score was categorized as healthy, intermediate, and unhealthy lifestyle based on the distribution of the unweighted lifestyle score [29].

### 2.7. Statistical Analysis 

IBM SPSS Statistics version 25.0 and S-PLUS 8.0 were used to perform all the analyses. Continuous variables were shown as mean and standard deviation (SD), and categorical variables were shown as numbers and percentages. The distribution difference of categorical variables between participants delivered by CS and participants delivered by vaginal delivery was compared by the Chi-square test. The associations of delivery mode with obesity risk, lifestyle with obesity risk, and the combination of delivery mode and lifestyle with obesity risk in children aged 6–18 years were evaluated by the binary logistic regression model. In all models, we adjusted for age, sex, region, ethnicity, and mother adiposity. Moreover, the effect-measure modification analysis was performed to evaluate the potential protective role of lifestyle on the effect of CS on childhood obesity with interaction on an additive scale using the S-PLUS 8.0 [30]. A 2-sided *p*-value < 0.05 was considered statistically significant.

## 3. Results

The characteristics of participants are showed in Table 1. A total of 44,961 students were enrolled into the current study, including 18,855 (41.9%) students delivered by CS and 26,106 (58.1%) students delivered by vaginal delivery. The mean age was younger in students delivered by CS than that in students delivered by vaginal delivery (9.8 vs. 11.2 years; *p* < 0.001). The distribution was significantly different for sex, region, nationality, healthy diet, regular physical activity, regular screen time, adequate sleep time, the number of healthy lifestyle factors, and mother adiposity between students delivered by CS and students delivered by vaginal delivery (*p* < 0.05). Moreover, students delivered by CS had a higher adiposity prevalence than that of those delivered by vaginal delivery (14.8% vs. 8.9%; *p* < 0.001). Only 2.7% of students kept 4 healthy lifestyle factors, 25.6% of students kept 1 of 4 healthy lifestyle factors, and 45.4% of students kept 2 of 4 healthy lifestyle factors. Most of the students (62.1%) kept an intermediate lifestyle (weighted lifestyle scores ranging 46.7–88.0), 23.7% of students kept an unhealthy lifestyle (weighted lifestyle scores ranging 0–46.6), and only 14.2% of students kept a healthy lifestyle (weighted lifestyle scores ranging 88–100). 

As shown in Table 2, compared with students delivered by vaginal delivery, students delivered by CS had a higher childhood adiposity risk (*OR* = 1.56; 95%*CI*: 1.47–1.67; *p* < 0.001) after adjustment for mother adiposity, age, sex, region, and ethnicity. Further adjustment for weighted lifestyle did not affect the associations (*OR* = 1.56; 95%*CI*: 1.46–1.66; *p* < 0.001). After stratification by age group, we observed similar associations.

Table 3 shows the association of lifestyle with obesity risk in children and adolescents. Compared with students keeping a healthy lifestyle, students keeping an unhealthy lifestyle had a substantially elevated risk of childhood obesity (*OR* = 1.30; 95%*CI*: 1.18–1.43; *p* < 0.001) after adjustment for potential confounders. Further adjustment of delivery mode resulted in an *OR* of 1.31 (95%*CI*: 1.19–1.44; *p* < 0.001). Additionally, stratification by age group did not modify the association in children aged 6–12 years.

When lifestyle groups and delivery mode were merged, there was a monotonic association between progressively unhealthy lifestyle factors and delivery mode (Figure 1). Compared with students delivered by vaginal delivery who kept a healthy lifestyle, students delivered by CS who kept an unhealthy lifestyle had a markedly enhanced obesity risk (*OR* = 2.06; 95%*CI*: 1.79–2.37; *p* < 0.001) in childhood and adolescence. Keeping a healthy lifestyle appears to lower the degree of the enhanced risk of obesity caused by CS but might not offset the enhanced risk of obesity caused by CS (*OR*= 1.59; 95%*CI*: 1.39–1.82). Moreover, we did not observe a significant interaction between lifestyle and delivery mode (*P*_interaction_ = 0.254), which suggested that the association of lifestyle with obesity risk was similar on the basis of delivery mode.

The association between lifestyle and risk of childhood obesity stratified by delivery mode is shown in Table 4. Among students delivered by vaginal delivery, both students with a healthy (*OR* = 0.76; 95%*CI*: 0.66–0.87; *p* < 0.001) and an intermediate lifestyle (*OR* = 0.79; 95%*CI*: 0.71–0.88; *p* < 0.001) had a lower risk of obesity than students with an unhealthy lifestyle. Among students delivered by CS, compared with students with an unhealthy lifestyle, both students with a healthy (*OR* = 0.77; 95%*CI*: 0.67–0.88; *p* < 0.001) and an intermediate (*OR* = 0.82; 95%*CI*: 0.74–0.92; *p* = 0.001) lifestyle also had a significantly lower risk of childhood obesity. We furthermore performed effect-measure modification analysis to evaluate the potential protective role of lifestyle on the effect of CS on childhood obesity and found no interaction on an additive scale (Relative Excess Risk due to Interaction, RERI = 0.18; 95%CI: −0.12, 0.49) (Appendix A).

## 4. Discussion

This study used the baseline survey data of a nationwide school-based program on healthy lifestyle interventions against childhood obesity and found that both delivery mode and lifestyle were markedly linked with risk of obesity in students aged 6–18 years. Students delivered by CS who maintained an unhealthy lifestyle had a higher risk of childhood obesity than those delivered by vaginal delivery who maintained a healthy lifestyle. We did not observe an interaction between delivery mode and lifestyle, and CS is associated with a higher risk of childhood obesity regardless of lifestyle. 

China has one of the highest CS rates in the world. In this study, we observed that the prevalence of CS in Chinese children aged 6–18 years is 41.9% (18,855/44,961), which was lower than that reported in Shanghai, China. Liu M and his colleagues used the 40,621 live births recorded between 2012 to 2014 and found that the CS rate was 45.1% in Eastern Branch of Shanghai First Maternity and Infant Hospital [31]. We speculated that the difference of the birth year and region might explain the divergence. The mean birth year of children in this study was 10 years earlier than that in Liu’s study. Moreover, the current study selected subjects from seven provinces across mainland China; however, Liu’s study just selected subjects from Shanghai based on a hospital. Thus, the CS rates reported in this study might be closer to the general CS rates in mainland China.

The increased risk of CS on childhood adiposity was similar to that found in a previous cohort study, which used the New Zealand birth cohort and found that planned CS was significantly associated with childhood adiposity (*RR* = 1.59; 95%*CI* 1.09–2.33) [32]. Although many previous studies have found that healthy lifestyle had a protective effect on childhood obesity risk [19,33,34], few studies determined the association between lifestyle and childhood obesity by a merged weighted lifestyle category. In the current study, we merged four main lifestyle factors linked with childhood obesity and computed a weighted score and observed that students who maintained an unhealthy lifestyle had 31% higher childhood obesity risk than those who maintained a healthy lifestyle. These findings suggested that maintaining an unhealthy lifestyle may elevate childhood obesity risk. 

To our knowledge, few studies have explored the association between merged delivery mode and lifestyle and risk of obesity in children and adolescents. Fang and his colleagues explored the association between merged polygenic risk score and lifestyle and childhood obesity risk and observed that children with high polygenic risk who maintained a healthy lifestyle had an 85% lower obesity risk than those with high polygenetic risk score who maintained an unhealthy lifestyle [35]. Nevertheless, whether a healthy lifestyle could offset the elevated childhood adiposity risk of CS was unknown. In this research, we used the baseline survey data of a large-scale national sample to investigate the association between CS and lifestyle and the risk of childhood obesity. 

The current research observed that CS significantly elevated the childhood obesity risk, but the elevated obesity risk might not be counteracted by a healthy lifestyle. Compared with children delivered by vaginal delivery, children delivered by CS had a 56% increased risk of childhood adiposity. Notably, if these children maintained a healthy lifestyle, they would reduce their risk of childhood adiposity by 23%, compared with those who maintained an unhealthy lifestyle. These findings suggested that the elevated risk of childhood adiposity caused by CS could be partly counteracted by a healthy lifestyle. It is a pity that children delivered by CS who maintained a healthy lifestyle still had a substantially elevated risk of childhood adiposity compared with children delivered by vaginal delivery who maintained a healthy lifestyle (*OR* = 1.59; 95%*CI*: 1.39–1.82). This finding indicated that the elevated risk of childhood adiposity caused by CS could not be counteracted by a healthy lifestyle. These results suggest that CS is an independent factor of childhood obesity. 

Although the current research utilized baseline survey data of a large-scale national school-based program on healthy lifestyle interventions against childhood obesity and found that the increased risk caused by CS could not be counteracted by a healthy lifestyle in childhood, several limitations of the current study should be mentioned below. Firstly, the association found in the current study was from a cross-sectional survey, which might limit us in making a causality. Nevertheless, the CS is an inherent factor of participants. Accordingly, we can make causal inferences between CS and childhood adiposity risk. Secondly, the data of delivery mode were retrospectively obtained from guardians of students. Consequently, the association in the current study might be caused by the existence of information bias. However, delivery mode was obtained mainly from their medical certificate of birth. Moreover, we also excluded participants with inconsistent self-reported delivery mode between baseline and 6 months’ follow-up surveys. Thirdly, though we adjusted for numerous potential confounders, the unmeasured confounders and modifiers remain. Fourthly, all the lifestyle factors in this study were acquired through a questionnaire. Thus, recall bias possibly persisted and resulted in non-differential misclassification, which would cause these associations toward the null hypothesis. Fifthly, seasonal variations in dietary consumption may occur because we have just collected information about dietary consumption in the previous week. Finally, the association observed in this study has not been repeated in other independent samples. 

## 5. Conclusions

In conclusion, the current research utilized baseline survey data of a nationwide school-based program on healthy lifestyle interventions against childhood obesity and found that CS and unhealthy lifestyle were markedly linked with the risk of obesity in children and adolescents aged 6–18 years. However, the increased risk caused by CS could not be counteracted by a healthy lifestyle in childhood. 

## Figures and Tables

**Figure 1 nutrients-14-03528-f001:**
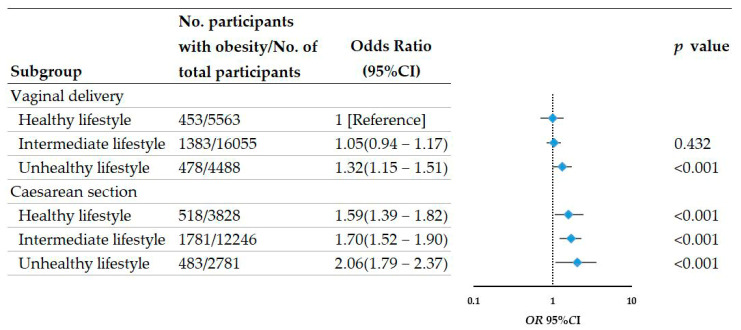
Risk of obesity according to delivery mode and lifestyle in children and adolescents aged 6–18 years.

**Table 1 nutrients-14-03528-t001:** Basic characteristics of study population in a study of the association between delivery and lifestyle with obesity in children and adolescents aged 6–18 years.

Characteristic	No. (%)			
	Total (*n* = 44,961)	CS Delivery (*n* = 18,855)	Vaginal Delivery (*n* = 26,106)	*p*-Value
Age, mean (SD), years	10.6 (3.2)	9.8 (3.0)	11.2 (3.2)	<0.001
Sex				<0.001
Male	22,580 (50.2)	9826 (52.1)	12,754 (48.9)	
Female	22,381 (49.8)	9029 (47.9)	13,352 (51.1)	
Region				<0.001
Rural	25,135 (55.9)	8728 (46.3)	16,407 (62.8)	
Urban	19,826 (44.1)	10,127 (53.7)	9699 (37.2)	
Nationality				<0.001
Han	40,873 (90.9)	17,279 (91.6)	23,594 (90.4)	
Others	4088 (9.1)	1576 (8.4)	2512 (9.6)	
Healthy lifestyle factors				
Healthy diet	13,993 (31.1)	5551 (29.4)	8442 (32.3)	<0.001
Regular physical activity	25,671 (57.1)	10,297 (54.6)	15,374 (58.9)	<0.001
Regular screen time	33,090 (73.6)	14,351 (76.1)	18,739 (71.8)	<0.001
Adequate sleep	14,730 (32.8)	6743 (35.8)	7987 (30.6)	<0.001
No. of healthy lifestyle factors				0.038
0	1724 (3.8)	712 (3.8)	1012 (3.9)	
1	11,524 (25.6)	4698 (24.9)	6826 (26.1)	
2	20,397 (45.4)	8615 (45.7)	11,782 (45.1)	
3	10,098 (22.5)	4306 (22.8)	5792 (22.2)	
4	1218 (2.7)	524 (2.8)	694 (2.7)	
Adiposity				<0.001
No	39,865 (88.7)	16,073 (85.2)	23,792 (91.1)	
Yes	5096 (11.3)	2782 (14.8)	2314 (8.9)	
Mother adiposity				<0.001
No	43,178 (96.0)	25,224 (96.6)	17,954 (95.2)	
Yes	1783 (4.0)	882 (3.4)	901 (4.8)	

Abbreviations: CS, Caesarean section; SD, standard deviation; No., number.

**Table 2 nutrients-14-03528-t002:** Association between delivery mode and obesity in children and adolescents aged 6–18 years.

Group	*n*	Model 1 ^a^		Model 2 ^b^	
		CS Delivery	*p* Value	CS Delivery	*p* Value
Total	44,961	1.56 (1.47–1.67)	<0.001	1.56 (1.46–1.66)	<0.001
Age group					
6–12 years	30,917	1.57 (1.46–1.68)	<0.001	1.56 (1.45–1.67)	<0.001
13–15 years	9702	1.86 (1.59–2.16)	<0.001	1.86 (1.60–2.17)	<0.001
16–18 years	4342	1.41 (1.10–1.80)	0.007	1.42 (1.11–1.81)	0.006

Abbreviations: CS, Caesarean section; Ref., reference; *OR*, odds ratio; *CI*, confidence interval. ^a^ Binary logistic regression model adjusted for age, sex, region, mother adiposity, and ethnicity. ^b^ Binary logistic regression model adjusted for model 1 and weighted lifestyle factors.

**Table 3 nutrients-14-03528-t003:** Association between lifestyle and obesity in children and adolescents aged 6–18 years.

Healthy Lifestyle Category	Model 1 ^a^			Model 2 ^b^		
	Healthy	Intermediate	Unhealthy	Healthy	Intermediate	Unhealthy
Total	1 [Ref.]	1.06 (0.99–1.15)	1.30 (1.18–1.43)	1 [Ref.]	1.06 (0.98–1.14)	1.31 (1.19–1.44)
*p* value		0.106	<0.001		0.147	<0.001
*p* value for trend	<0.001			<0.001		
Age group						
6–12 years	1 [Ref.]	1.07 (0.97–1.17)	1.32 (1.18–1.48)	1 [Ref.]	1.06 (0.96–1.16)	1.32 (1.18–1.48)
*p* value		0.172	<0.001		0.243	<0.001
*p* value for trend	<0.001			<0.001		
13–15 years	1 [Ref.]	1.02 (0.85–1.21)	1.04 (0.82–1.31)	1 [Ref.]	1.01 (0.85–1.21)	1.07 (0.85–1.36)
*p* value		0.875	0.761		0.887	0.558
*p* value for trend	0.397			0.526		
16–18 years	1 [Ref.]	0.93 (0.74–1.26)	1.23 (0.87–1.75)	1 [Ref.]	0.96 (0.74–1.26)	1.27 (0.89–1.81)
*p* value		0.782	0.242		0.783	0.181
*p* value for trend	0.379			0.003		

Abbreviations: Ref., reference; *OR*, odds ratio; *CI*, confidence interval. ^a^ Binary logistic regression model adjusted for age, sex, region, mother adiposity, and ethnicity. ^b^ Binary logistic regression model adjusted for model 1 and delivery mode.

**Table 4 nutrients-14-03528-t004:** Risk of obesity according to healthy lifestyle category within delivery mode in children and adolescents aged 6–18 years.

Healthy Lifestyle Category	Vaginal Delivery			CS Delivery		
	Healthy	Intermediate	Unhealthy	Healthy	Intermediate	Unhealthy
Total	0.76 (0.66–0.87)	0.79 (0.71–0.88)	1 [Ref.]	0.77 (0.67–0.88)	0.82 (0.74–0.92)	1 [Ref.]
*p* value	<0.001	<0.001		0.001	<0.001	
*p* value for trend	<0.001			<0.001		
Age group						
6–12 years	0.78 (0.66–0.92)	0.78 (0.69–0.89)	1 [Ref.]	0.75 (0.65–0.88)	0.82 (0.73–0.93)	1 [Ref.]
*p* value	<0.001	0.004		<0.001	0.002	
*p* value for trend	0.002			<0.001		
13–15 years	0.77 (0.57–1.05)	0.89 (0.69–1.15)	1 [Ref.]	1.23 (0.84–1.82)	1.05 (0.74–1.48)	1 [Ref.]
*p* value	0.095	0.377		0.279	0.789	
*p* value for trend	0.026			<0.001		
16–18 years	0.80 (0.53–1.21)	0.72 (0.50–1.04)	1 [Ref.]	0.76 (0.37–1.53)	0.82 (0.43–1.56)	1 [Ref.]
*p* value	0.293	0.082		0.643	0.824	
*p* value for trend	0.411			0.466		

Abbreviations: CS, Caesarean section; Ref., reference; *OR*, odds ratio; *CI*, confidence interval.

## Data Availability

The raw data supporting the conclusions of this article will be made available by the authors, without undue reservation.

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
