# Peer review of "Does a Healthy Lifestyle Lower the Elevated Risk of Obesity Caused by Caesarian Section Delivery in Children and Adolescents?"

_nutrients, 2022, doi:10.3390/nu14173528_

Round 1

Reviewer 1 Report

·         The title “Healthy lifestyle lower the elevated risk of childhood obesity caused by caesarian section delivery” and the abstract’s conclusion “Adherence to a favorable lifestyle did not counteract the elevated risk of childhood obesity caused by CS” are not consistent. It sounds like the title is misleading and different from the study’s conclusion on page 8, lines 303-305 “CS and unhealthy lifestyle were independently associated with the childhood obesity risk. Moreover, the increased risk caused by CS might not be offset by a favorable lifestyle in childhood.”

Page 3, lines 103-105: “Dietary consumption, including vegetables, fruits, meats, dairy, sugar-sweetened beverages, fried foods, and desserts was reported by students or their guardians. They reported the frequency (day) and amount (servings) over the past a week (7 days).”

·         It is unclear why the authors used 7-day food record and not a food frequency questionnaire (FFQ). FFQ can capture more comprehensive information on usual dietary intake, including seasonal variations. Please explain.

Page 3, lines 140-142: “Sleep duration was defined as adequate sleep and inadequate sleep according to the recommendation of the Canadian sedentary behavior guidelines for children and youth29.”

·         Were the Canadian sedentary behaviour guidelines for children and youth validated for Chinese children or tested in a pilot study previously?

·         How was “mother obesity” defined? Was it based on pre-pregnancy or pregnancy BMI or something else?

·         In tables 2, 3 and 4, why did the authors classify age as 6-9, 10-13, and 14-17? What was the rationale for that?

·         Authors could perform effect-measure modification analysis to evaluate the potential protective role of lifestyle on the effect of CS on childhood obesity. This statistical analysis calculates the relative risk for interaction (RERI). You can learn more about this method in this paper: Knol MJ, van der Tweel I, Grobbee DE, Numans ME, Geerlings MI. Estimating interaction on an additive scale between continuous determinants in a logistic regression model. Int J Epidemiol 2007;5: 1111-1118.

·         Please add the questionnaire for calculating healthy lifestyle score as the supplementary file.

Author Response

Point 1: The title “Healthy lifestyle lower the elevated risk of childhood obesity caused by caesarian section delivery” and the abstract’s conclusion “Adherence to a favorable lifestyle did not counteract the elevated risk of childhood obesity caused by CS” are not consistent. It sounds like the title is misleading and different from the study’s conclusion on page 8, lines 303-305 “CS and unhealthy lifestyle were independently associated with the childhood obesity risk. Moreover, the increased risk caused by CS might not be offset by a favorable lifestyle in childhood.”

Response 1: Thank you for your comments. We have replaced the title with “Is a healthy lifestyle lowering the elevated risk of childhood obesity caused by caesarian section delivery in children and adolescents?” in the revised manuscript. Please refer to the Title in the revised manuscript, page 1.

Point 2: Page 3, lines 103-105: “Dietary consumption, including vegetables, fruits, meats, dairy, sugar-sweetened beverages, fried foods, and desserts was reported by students or their guardians. They reported the frequency (day) and amount (servings) over the past a week (7 days).” It is unclear why the authors used 7-day food record and not a food frequency questionnaire (FFQ). FFQ can capture more comprehensive information on usual dietary intake, including seasonal variations. Please explain.

Response 2: Information of Dietary consumption in the current study is from the baseline survey of a nationwide school-based health lifestyles interventions in Chinese children against obesity project. In this survey, a 7-day food frequency questionnaire was used to collect the dietary consumption information. Thus, seasonal variations might be a limitation, which has been mentioned in the limitation section. Please refer to lines 98 and 278-280, pages 3 and 8.

Point 3: Page 3, lines 140-142: “Sleep duration was defined as adequate sleep and inadequate sleep according to the recommendation of the Canadian sedentary behavior guidelines for children and youth29.”. Were the Canadian sedentary behaviour guidelines for children and youth validated for Chinese children or tested in a pilot study previously?

Response 3: According to the Notice of the Chinese General Office of the Ministry of Education on Further Strengthening the Sleep Management of Primary and Middle School Students, adequate sleep was defined if sleeping time ≥10 hours for primary students, ≥9 hours for the middle students and ≥8 hours for the high middle students, which was consistent with the recommendation of the Canadian sedentary behavior guidelines for children and youth. We have changed the statement. Please refer to lines 124-127, page 3.

Point 4: How was “mother obesity” defined? Was it based on pre-pregnancy or pregnancy BMI or something else?

Response4: Mother obesity was defined based on maternal BMI at the time of the survey. Please refer to lines 111-112, page 3.

Point 5:  In tables 2, 3 and 4, why did the authors classify age as 6-9, 10-13, and 14-17? What was the rationale for that?

Response 5: The subjects of this study are between 6-18 years old. According to your comment, we had re-grouped them into three age groups: primary school students (6-12 years old), middle school students (13-15 years old) and high school students (16-18 years old), and found that the associations were similar to the previous group. Please refer to Table 2,3 and 4 in the revised manuscript.

Point 6: Authors could perform effect-measure modification analysis to evaluate the potential protective role of lifestyle on the effect of CS on childhood obesity. This statistical analysis calculates the relative risk for interaction (RERI). You can learn more about this method in this paper: Knol MJ, van der Tweel I, Grobbee DE, Numans ME, Geerlings MI. Estimating interaction on an additive scale between continuous determinants in a logistic regression model. Int J Epidemiol 2007;5: 1111-1118.

Response 6: Thank you for your valuable suggestion. According to your comments, we have performed the effect-measure modification analysis to evaluate the potential protective role of lifestyle on the effect of CS on childhood obesity, and found that no interaction on an additive scale (RERI=0.18; 95%CI: -0.12, 0.54). Please refer to Table S1 in supplemental materials and lines 206-209, page 6.

Table S1. Output of logistic regression model with delivery mode and lifestyle as dichotomous determinants and product of delivery and lifestyle entered into the model. Outcome is obesity.

Parameters

Estimate

SE

95%CI of OR

OR

Lower

Upper

Delivery

0.587

0.033

1.80

1.69

1.92

Lifestyle

0.250

0.054

1.28

1.16

1.43

Delivery × Lifestyle

-0.020

0.065

0.80

0.84

1.14

Constant

-2.377

0.024

RERI(95%CI)

0.18 (-0.12, 0.49)

Abbreviations: SE, Standard Error; OR, Odds Ratio; RERI, Relative Excess Risk due to Interaction.

Reviewer 2 Report

The authors of the research entitled " Healthy lifestyle lower the elevated risk of childhood obesity  caused by caesarian section delivery” presented in a very interesting way the problem of childhood obesity, which can lead to many metabolic diseases. It is extremely important to search for all possible connections, including the type of childbirth, in order to be able to take preventive measures as soon as possible to protect the health of children.

Nevertheless, they did not avoid mistakes:

- please check all abbreviations and their descriptions used in the tables.- tables and figures should be cited uniformly throughout the article

-please explain what are the strengths and limitations of your study?

- references list –if possible, add doi number

- please adjust the citation method to the journal's requirements

After all corrections have been made, the manuscript should be published.

Author Response

Point1: The authors of the research entitled " Healthy lifestyle lower the elevated risk of childhood obesity caused by caesarian section delivery” presented in a very interesting way the problem of childhood obesity, which can lead to many metabolic diseases. It is extremely important to search for all possible connections, including the type of childbirth, in order to be able to take preventive measures as soon as possible to protect the health of children.

Nevertheless, they did not avoid mistakes:

Response 1: Thank you for your valuable comments, we have carefully changes all these mistakes in the revised manuscript. Please refer to revised manuscript.

Point 2: please check all abbreviations and their descriptions used in the tables.- tables and figures should be cited uniformly throughout the article

Response 2: We have check all abbreviations in tables and figures and improved them. Please refer to lines 17, 37, 162, 170, 183 and Figure 1.

Point 3: please explain what are the strengths and limitations of your study?

Response 3: We have furthermore clarified the strengths and limitations of the study in the Discussion section. Please refer to lines 263-281, pages 7-8.

Point 4: references list –if possible, add doi number

Response 4: We have added their doi for all references except 24 that do not have a doi.

Point 5: please adjust the citation method to the journal's requirements

 Response 5: We have edited the citation method according to the journal’s requirements.
